# Landsat-Based Estimation of Seasonal Water Cover and Change in Arid and Semi-Arid Central Asia (2000–2015)

**Xianghong Che** [1] , **Min Feng** [2,3,*], **Joe Sexton** [3], **Saurabh Channan** [3], **Qing Sun** [4], **Qing Ying** [5], **Jiping Liu** [1] **and Yong Wang** [1]

[1]  Research Center of Government Geographic Information System, Chinese Academy of Surveying & Mapping, Beijing 100830, China; chexh.15b@gmail.com (X.C.); liujp@casm.ac.cn (J.L.); wangyong@casm.ac.cn (Y.W.)

[2]  Institute of Tibetan Plateau Research, Chinese Academy of Sciences, Beijing 100101, China

[3]  terraPulse Inc., North Potomac, MD 20878, USA; sexton@terrapulse.com (J.S.); schannan@umd.edu (S.C.)

[4]  Collaborative Innovation Center on Forecast and Evaluation of Meteorological Disasters (CIC-FEMD), Nanjing University of Information Science & Technology, Nanjing 210044, China; sunqing@nuist.edu.cn or sunqingmeteo@gmail.com

[5]  Department of Geographical Sciences, University of Maryland, College Park, MD 20742, USA; amy.qing.ying@gmail.com or qying@umd.edu

\*  Correspondence: mfeng@itpcas.ac.cn

**Abstract:** Surface water is of great importance to ecosystems and economies. Crucial to understanding hydrological variability and its relationships to human activities at large scales, open-access satellite datasets and big-data computational methods are now enabling the global mapping of the distribution and changes of inland water over time. A machine-learning algorithm, previously used only to map water at single points in time, was applied over 16 years of the USGS Landsat archive to detect and map surface water over central Asia from 2000 to 2015 at a 30-m, monthly resolution. The resulting dataset had an overall classification accuracy of 99.59% (±0.32% standard error), 98.24% (±1.02%) user's accuracy, and 87.12% (±3.21%) producer's accuracy for water class. This study describes the temporal extension of the algorithm and the application of the dataset to present patterns of regional surface water cover and change. The findings indicate that smaller water bodies are dramatically changing in two specific ecological zones: the Kazakh Steppe and the Tian Shan Montane Steppe and Meadows. Both the maximum and minimum extent of water bodies have decreased over the 16-year period, but the rate of decrease of the maxima was double that of the minima. Coverage decreased in each month from April to October, and a significant decrease in water area was found in April and May. These results indicate that the dataset can provide insights into the behavior of surface water across central Asia through time, and that the method can be further developed for regional and global applications.

**Keywords:** surface water; Landsat; time series detection; central Asia

## 1. Introduction

Inland surface water bodies including natural fresh and saline lakes, rivers, and artificial reservoirs cover a small (~3%) and dynamic portion of global land area, and are crucial to the well-being of humans and other living organisms [1,2]. In central Asia and other similar arid or semi-arid areas, the sparse spatio-temporal distribution of surface water results in regional vulnerability to variation caused by climate and human activities [3–6]. Accurately detecting and monitoring water bodies in

this area and analyzing their changes over time can provide insights into the region's water resources as well as their causes and consequences in natural and agricultural ecosystems.

In recent decades, the spatio-temporal extent of archived satellite images has been used to map the distribution of surface water as a static, often binary variable [7–11]. Optical sensors with coarse spatial resolution and daily orbits, for example, the Advanced Very-High-Resolution Radiometer (AVHRR), Moderate Resolution Imaging Spectroradiometer (MODIS), and Visible Infrared Imaging Radiometer Suite (VIIRS), can detect water-body changes at a high temporal resolution. Klein et al. [12] used AVHRR and MODIS data to estimate the monthly extents of inland water bodies from 1986 to 2012 over central Asia. Sun et al. [13] used 8-day MODIS composite data from 2000 to 2010 to map water surface changes across 629 lakes in China. However, the relatively coarse (> 100 m) resolutions of these data limit their capability for monitoring small lakes.

To understand the dynamics of small lakes, medium-resolution (e.g., Landsat) satellite sensors are beginning to be employed to map regional or global surface water. Feng et al. [1] presented a machine learning-based approach for producing a global, circa-2000 inland surface water body dataset based on atmospherically corrected surface reflectance estimates, topographic indices, and prior coarse-resolution water layers. Yamazaki et al. [14] developed a water frequency index and multi-scene mean indexes (i.e., reflectance, NDWI (Normalized-Difference Water Index), NDVI (Normalized-Difference Vegetation Index), and brightness temperature) by using multi-temporal Landsat images [15], and separated permanent water bodies from temporal water-covered areas from 1990 to 2010 by exploring different thresholds for these indexes. Based on seasonally continuous Landsat TM (Thematic Mapper) /ETM+ (Enhanced Thematic Mapper Plus) data with a cloud coverage < 50%, Tulbure et al. [16] used generic random forest-based models with large volume of water/non-water pixels selected via the visual interpretation of Landsat images to synoptically map the extent and dynamics of surface water and flooding (1986–2011) over the Murray–Darling Basin. Mueller et al. [17] applied a regression model with manually selected training data to map the surface water from 25 years of Landsat imagery across Australia. Halabisky et al. [18] developed a four endmember spectral mixture analysis (SMA) model of a time series of Landsat satellite imagery to measure the sub-pixel wetland inundation of 750 wetlands in USA from 1984 to 2011, where different land cover endmembers were directly selected from each Landsat image. However, these surface water datasets either had irregular time periods or the water/non-water samples required manual collection, thus limiting their capability of precisely delineating the surface water dynamics of a long time-series in large regions. Pekel et al. [19] retrieved a high-resolution mapping dataset of global surface water quantifying changes in the global surface water over the past 32 years from 1984 to 2015 at a 30-m resolution using the entire archive of Landsat 5 TM, Landsat 7 ETM+, and Landsat 8 OLI images. Each pixel was individually classified into water /non-water using an expert system and a set of global samples. While this dataset can provide some regional insights, its global calibration limits the certainty of its observations at a regional or local scale [19].

Satellite-based studies of lake dynamics in central Asia have been limited to specific lakes and/or irregular time periods, with little knowledge of their accuracy [20–23]. To improve our understanding of surface water dynamics at a regional scale, this study attempts to produce a regionally accurate, long-term water-body dataset with a 30-m, monthly resolution to study decadal and seasonal changes. Here, the study describes the temporal extension of the algorithm and its first application in central Asia.

## 2. Materials and Methods

### 2.1. Study Area

The study area included a large portion of central Eurasia, covering the former Soviet republics of Kazakhstan, Kyrgyzstan, Tajikistan, Turkmenistan, Uzbekistan, and the Xinjiang Autonomous Territory of China. The basin of the Caspian Sea was excluded to focus on small water bodies in the remaining portion of central Asia [24] (Figure 1). Elevation in the study area increases to the east from the coast

of the Caspian Sea in western Turkmenistan and Kazakhstan to the mountainous terrain in eastern Kazakhstan and Uzbekistan and across Kyrgyzstan and Tajikistan. This region has a continental climate with cold winters and hot summers, increasing temperature and precipitation from north to south, and increasing precipitation and decreasing temperature with elevation [25]. Major land cover types include grasslands, deserts, and irrigated agriculture. As one of the world's largest closed drainage basins, the distribution of water bodies in this region is uneven, and is comprised of many large water bodies throughout, with numerous natural lakes and artificial water reservoirs concentrated at the northwest grassland of Kazakhstan and the southern mountain region [26,27]. Although water bodies in this region are of high ecological and economic importance, significant changes have occurred in the last decades, thus leading to land desertification, salinization, the degradation of vegetation, and biodiversity loss [3,12]. Accurate datasets and time-series records of water bodies are helpful for not only expanding our understanding of natural variability and human interaction, but also addressing these ecosystem and environmental issues.

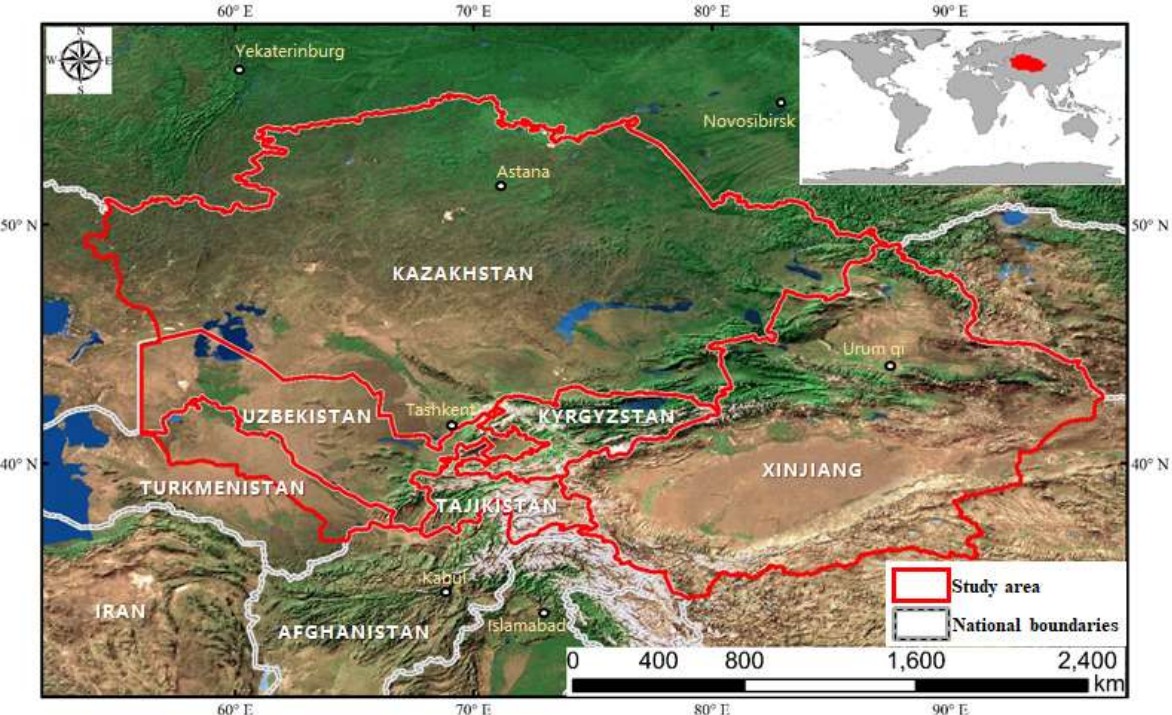

**Figure 1.** Extent of the study area (Kazakhstan, Uzbekistan, Turkmenistan, Kyrgyzstan, Tajikistan, and the Xinjiang Autonomous Territory of China, excluding the Caspian Sea basin).

*2.2. Datasets*

2.2.1. Landsat Data

Level-1 Terrain Corrected (L1T) Landsat images between 2000 and 2015 with a cloud cover < 80% were selected. The study area was covered by 315 Landsat World Reference System (WRS)-2 scenes, and 70,186 Landsat images were downloaded from USGS (U.S. Geological Survey)/EROS (Earth Resources Observation Satellite) [28] (Figure 2). These included 18,294 Landsat-5 TM images, 39,982 Landsat-7 ETM+ images, and 11,910 Landsat-8 Operational Land Imager (OLI) images. Eastern Xinjiang had a greater sampling intensity due to repeated acquisitions of Landsat data from international ground stations from multiple regions (i.e., China, central Asia, and India) [29]. Each scene was covered by a minimum of 149 and a maximum of 487 Landsat images, with an average of 19 images for each year. Fewer images (~9/year) were available in the northern part of the study area, although the overlap between adjacent orbits is denser at higher latitudes [30].

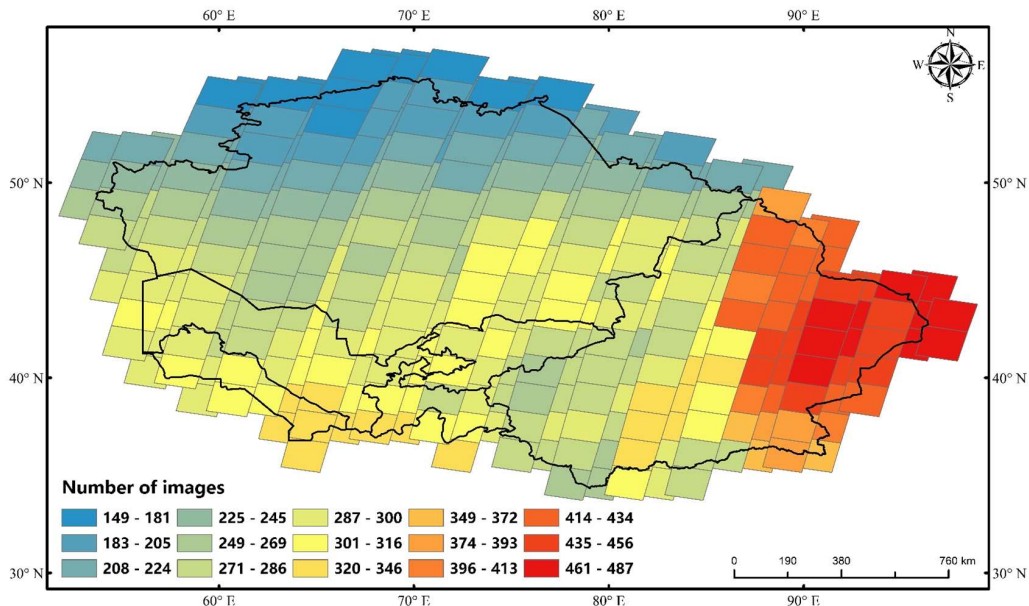

**Figure 2.** Coverage of the number of Landsat 5 TM, Landsat 7 ETM+, and Landsat 8 OLI images from 2000 to 2015.

The Landsat images were processed for surface reflectance using the Landsat Ecosystem Disturbance Adaptive Processing System (LEDAPS) [31,32] for TM/ETM+ images and the Landsat Surface Reflectance Code (LaSRC) [32–34] for OLI images. Fmask [35] was applied to identify the cloud and cloud shadow in each Landsat image.

### 2.2.2. Ancillary Datasets

The 30-m resolution ASTER Global Digital Elevation Model (GDEM) v2.0 was downloaded from [36]. Slope and hill-shade were calculated using GDAL DEM [37].

The 250-m resolution MODIS water dataset (MOD44W) [38] was used as a reference layer for water sampling with a water detection method from the Landsat images. The dataset was produced from MODIS 250-m reflectance data and elevation estimates from the Shuttle Water Body Dataset (SWBD) [39]. The global water mask was downloaded from the Global Land Cover Facility (GLCF) website [40].

The Joint Research Centre (JRC) Yearly and Monthly Water Classification History (v1.0) dataset was used to evaluate the detection performance [18]. This dataset quantified the changes in global surface water over the past 32 years from 1984 to 2015 at a 30-m resolution by using the entire archive of the Landsat 5 TM, Landsat 7 ETM+, and Landsat 8 OLI images. Each pixel was individually classified into water/non-water by using an expert system and the results were collated into a yearly and monthly history, which were download from the Google Earth engine [41].

### 2.3. Methods

### 2.3.1. Water Detection Algorithm

Surface water was identified in each Landsat image by using the algorithm developed by Feng et al. [1]. The algorithm automatically identifies water cover pixels in each Landsat image by building a locally optimized decision tree model. For each Landsat image, possible water pixels with varying certainty were identified by examining the multispectral water and topographic indices (e.g., NDWI and MNDWI (Modified Normalized Difference Water Index)). These strata were compared to a reference water layer (e.g., the MODIS water mask dataset) to establish additional sampling strata, representing different certainties of being water and non-water in the image. The sample was randomly collected in each stratum, but assigned different weights where samples in the high certainty stratum

were assigned higher weights than those in the low certainty stratum. A decision tree model was built from the points using the C5 algorithm [42], then the model was applied to the image to identify water in the Landsat image. A detailed description of the algorithm is described in Feng et al. [1].

### 2.3.2. Temporal Interpolation

Landsat images were distributed in the WRS-2 scenes and the Universal Transverse Mercator (UTM) projection zone corresponding to the scene. Landsat images in each scene vary in spatial extent and overlap with images from neighboring scenes, thus making it difficult to directly extract a time-series for temporal analysis [43]. To provide a consistent spatiotemporal reference system and facilitate temporal analysis, the region was divided into 668 grids with MODIS Sinusoidal Projection [44], and each grid consisted of 1000 × 1000 90-m pixels (Figure 3). The water dataset was re-projected to the grid projection using the nearest-neighbor interpolation.

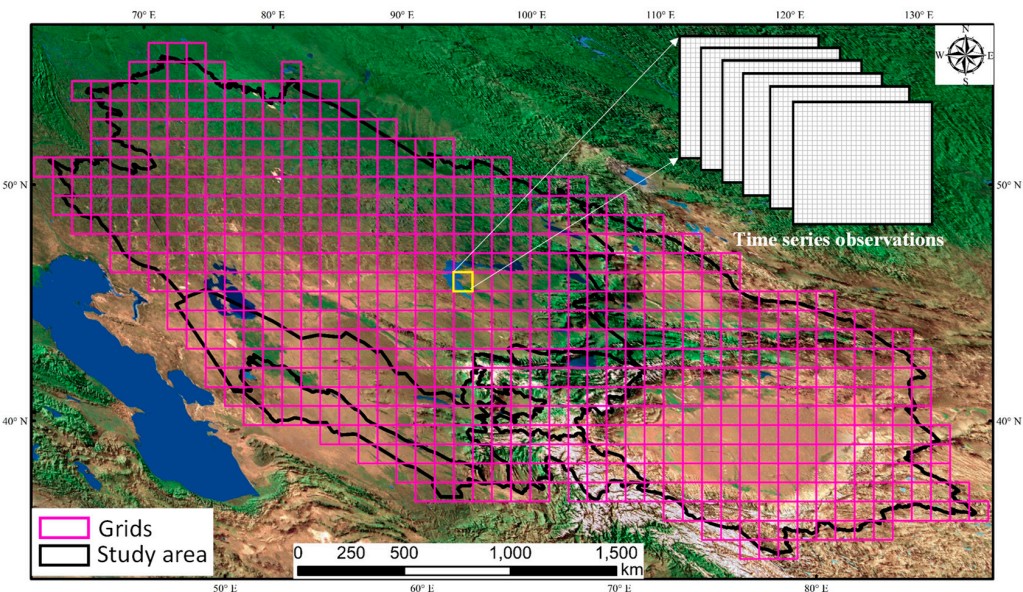

**Figure 3.** The grid distribution of the study area.

The irregular time-series of directly observed water records were linearly interpolated to daily frequency and then aggregated to monthly estimates for analysis. Based on the hypothesis that surface water changes linearly during the time interval, the dates with direct (image) observations were assigned 1.0 if water was identified at the pixel for the date, otherwise 0.0. Dates between direct estimates were linearly interpolated ranging from 0 to 1, then the interpolated results in a month were averaged. The mapped result for the month at the pixel was assigned as water if the averaged value was greater than 0.5, otherwise land. An example of the interpolation is presented in Figure 4.

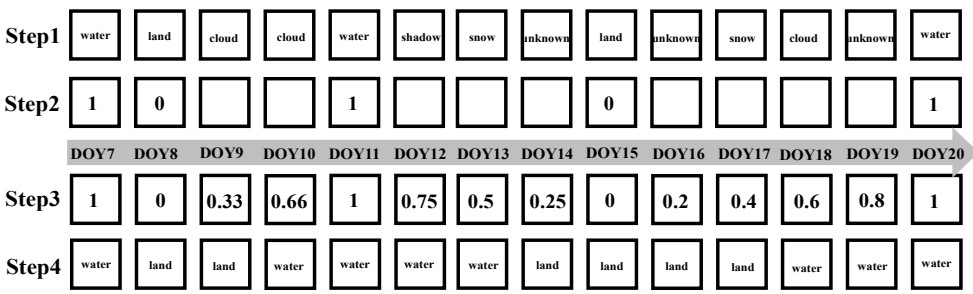

**Figure 4.** An example of the temporal interpolation of daily water estimates for a pixel.

### 2.3.3. Implementation and Data Processing

The method was implemented in C and Python. Open-source libraries GDAL/OGR [45], PROJ4 [46], NumPy [47], SciPy [48], and Matplotlib [49] were used in the implementation.

Data processing was carried out on clusters at the Global Land Cover Facility (GLCF), University of Maryland. Due to the heavy data processing requirement, 30 servers (each server had 8-32 cores and 64 to 128 Gigabytes of memory) were allocated for the task. The code was executed in parallel across scenes to use the multiple cores on each server. Identifying water in all of the selected Landsat images took ~2 weeks, followed by another ~2 weeks for temporal interpolation and analysis.

### 2.3.4. Accuracy Assessment

The interpreted samples were used as reference data to validate the water mapping results. The overall accuracy and water/non-water user's accuracy and producer's accuracy were estimated [50]. A reference sample was collected through a stratified random design to represent the range of different water conditions and seasonality in the area. All Landsat images between 2010 and 2015 were processed for water detection and then composited to estimate the percentage of water-occurrence frequency (WOF) among all of the valid observations at each 90-m resolution pixel in the study area. The study area was then divided into three strata: permanent water (WOF >= 95%), seasonal water (7 < WOF < 95%), and permanent land (WOF <= 7%) (Figure 5), and 200 points were randomly collected in each of the three strata. The 600 points were then interpreted as "water" or "non-water" in each season (spring, summer, autumn, and winter) by visually examining the Landsat images. In order to reduce the interpretation work load, each season was randomly selected from the six years (2010 to 2015) for interpretation at a selected point.

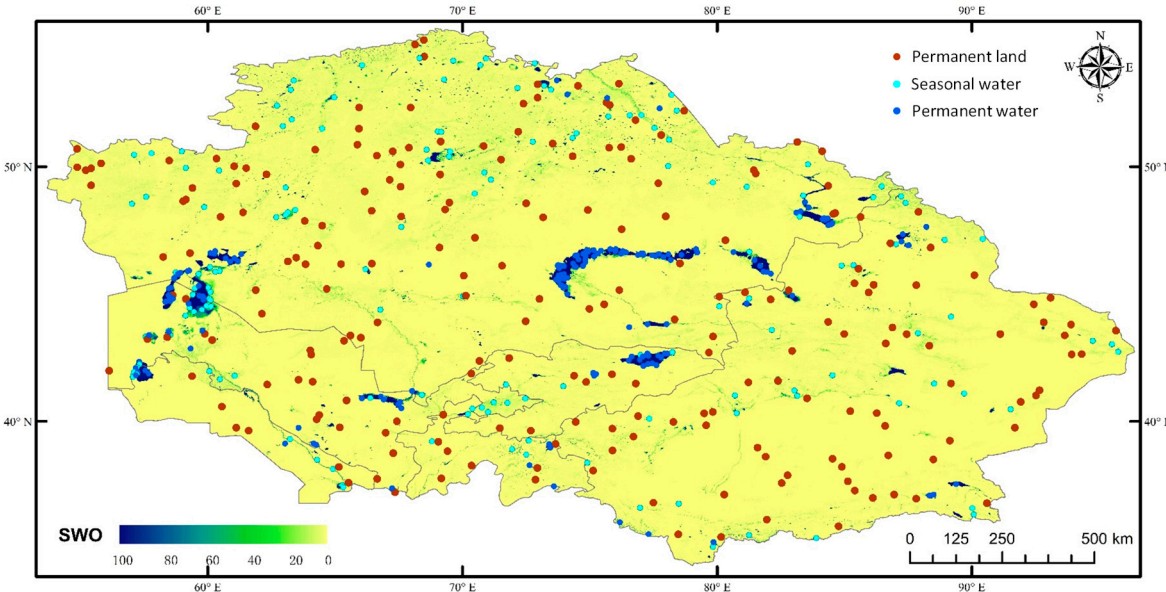

**Figure 5.** The spatial distribution of the collected samples for validation.

The Landsat-derived water records were extracted for the selected points and compared to the corresponding visual interpretations. The sampling probability for the strata were applied as weights for unbiased estimation of a regional area. Based on a Taylor series approximation, standard errors were also calculated for the accuracy metrics to measure the reliability of the accuracy estimates [51].

## 3. Results

### 3.1. Surface Water Estimates and Its Uncertainties

A sample of 2356 pixels was selected in the study area and included 780 in permanent water, 792 in seasonal water, and 784 in permanent land. The overall classification accuracy was 99.59% (standard error, SE = ± 0.32%); the user and producer accuracies of surface water in time were 87.12% (SE of ± 3.21%) and 98.24% (SE ± 1.02%), respectively (Table 1). The commission rate of non-water reference pixels to the estimated pixel class was 12.88%, and the commission rate of water pixels to the modeled non-water water class was 1.76%. Visual interpretation of the validation samples revealed that commission errors were strongly associated with undetected cloud and terrain shadows and the misclassification of snow and ice as water.

**Table 1.** The results of the accuracy and corresponding variance of water extraction from 2010 to 2015.

| | | Reference | | |
| | | Water | Non-Water | UA |
|---|---|---|---|---|
| | Water | 30.246 | 1.323 | 87.12 (± 3.21) |
| **Map (Algorithm)** | Non-water | 0.412 | 68.005 | 99.96 (± 0.04) |
| | PA | 98.24 (±1.02) | 99.15 (±0.59) | 99.59 (±0.32) |

Accuracy was high across the four seasons with the lowest OA (> 85.36%) in autumn and winter, followed by OA (> 92.92%) in spring, and the highest OA (> 93.67%) in summer. The accuracy of seasonal water was lower (OA = 85.36 – 93.67%) than that of permanent water (> 99.01%) or land (> 98.78%) in all seasons. The producer's accuracy (PA = 87.23 – 99.12%) was higher than the user's accuracy (< 82.58%) in seasonal water, suggesting over-estimation of surface-water coverage in seasonal water bodies, in other words, a bias toward water over non-water detection (Table 2).

**Table 2.** Accuracy of instantaneous water detection in seasonal inundation classes.

| Strata | | Class | Spring | Summer | Autumn | Winter |
|---|---|---|---|---|---|---|
| | OA | | 99.03 (±0.02) | 100 (±0.00) | 99.52 (±0.02) | 99.01 (±0.03) |
| **Permanent** | UA | Water | 98.87 (±1.01) | 100 (±0.00) | 100 (±0.00) | 99.57 (±0.30) |
| **water** | | Non-water | 100 (±0.00) | 100 (±0.00) | 92.12 (±0.03) | 97.89 (±0.02) |
| **N = 780** | PA | Water | 100 (±0.00) | 100 (±0.00) | 99.42 (±0.54) | 99.23 (±1.10) |
| | | Non-water | 95.02 (±0.02) | 100 (±0.00) | 100 (±0.00) | 98.83 (±0.02) |
| | OA | | 92.92 (±0.13) | 93.67 (±0.11) | 90.11 (±0.11) | 85.36 (±0.19) |
| **Seasonal** | UA | Water | 74.16 (±3.14) | 82.58 (±2.77) | 72.87 (±2.64) | 70.21 (±1.99) |
| **water** | | Non-water | 97.64 (±0.08) | 98.83 (±0.03) | 98.12 (±0.05) | 99.92 (±0.01) |
| **N = 792** | PA | Water | 87.23 (±2.99) | 94.45 (±1.38) | 90.01 (±2.01) | 99.12 (±0.42) |
| | | Non-water | 93.87 (±0.11) | 93.12 (±0.12) | 89.67 (±0.12) | 83.49 (±0.17) |
| | OA | | 99.87 (± 0.36) | 99.99 (±0.02) | 99.59 (±0.82) | 98.78 (±3.56) |
| **Permanent** | UA | Water | 92.32 (±10.89) | 98.48 (±4.01) | 90.26 (±16.32) | 80.89 (±19.32) |
| **land** | | Non-water | 98.71 (±0.14) | 98.56 (±0.11) | 97.43 (±0.22) | 99.78 (±0.03) |
| **N = 784** | PA | Water | 100 (±0.00) | 100 (±0.00) | 100 (±0.00) | 100 (±0.00) |
| | | Non-water | 99.89 (±0.49) | 99.98 (±0.02) | 99.56 (±0.89) | 98.01 (±0.83) |

Errors were more common in winter, suggesting the misclassification of snow and ice as liquid water. The surface water map on a single date tended to be contaminated by cloud, cloud shadow, and gaps. Using temporal interpolation, these issues were able to be fixed on the monthly water map, which indicated the good performance of the interpolation (Figure 6).

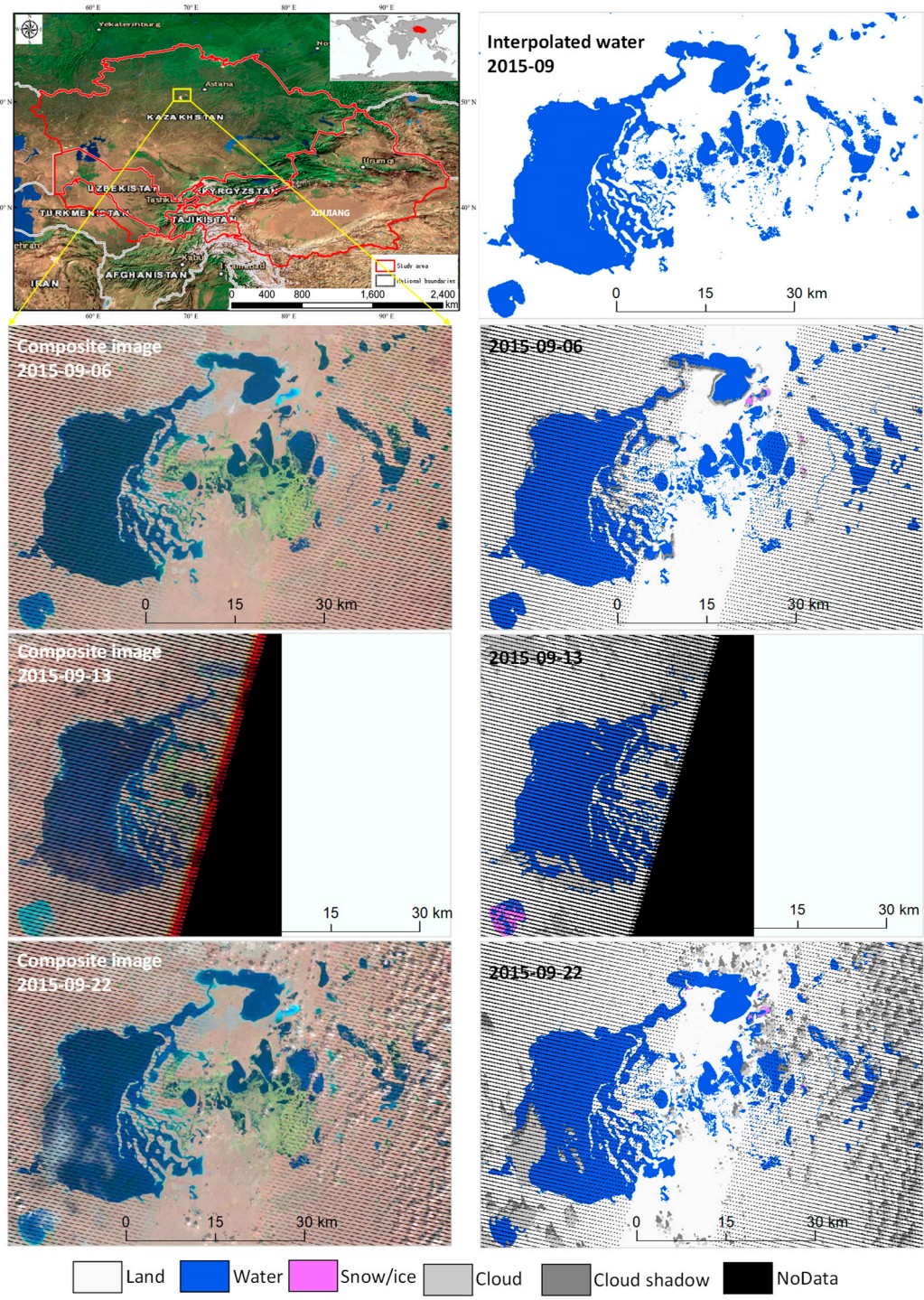

**Figure 6.** The water map directly derived from the available Landsat images and interpolated water map. The zoomed-in areas are from Tengiz-Korgalzhyn Lake in the north of Kazakhstan.

*3.2. Surface Water Dynamics in the Area*

Given the confusion of water with snow and ice, the analysis of surface-water dynamics was restricted to non-winter seasons. The 10-month inundation frequency from April to October between 2000 and 2015 was overlaid with terrestrial ecological zones in central Asia where 24 ecological zones [52] were aggregated as eight terrestrial ecological zones by combining adjacent regions with similar biotypes (Figure 7). The overlay showed that larger water bodies were distributed in middle latitude semi-desert areas with a high inundation frequency. The smaller water bodies were mainly

located in the Kazakh Steppe and Tian Shan Montane Steppe and Meadows ecoregions were also associated with high variability in water cover.

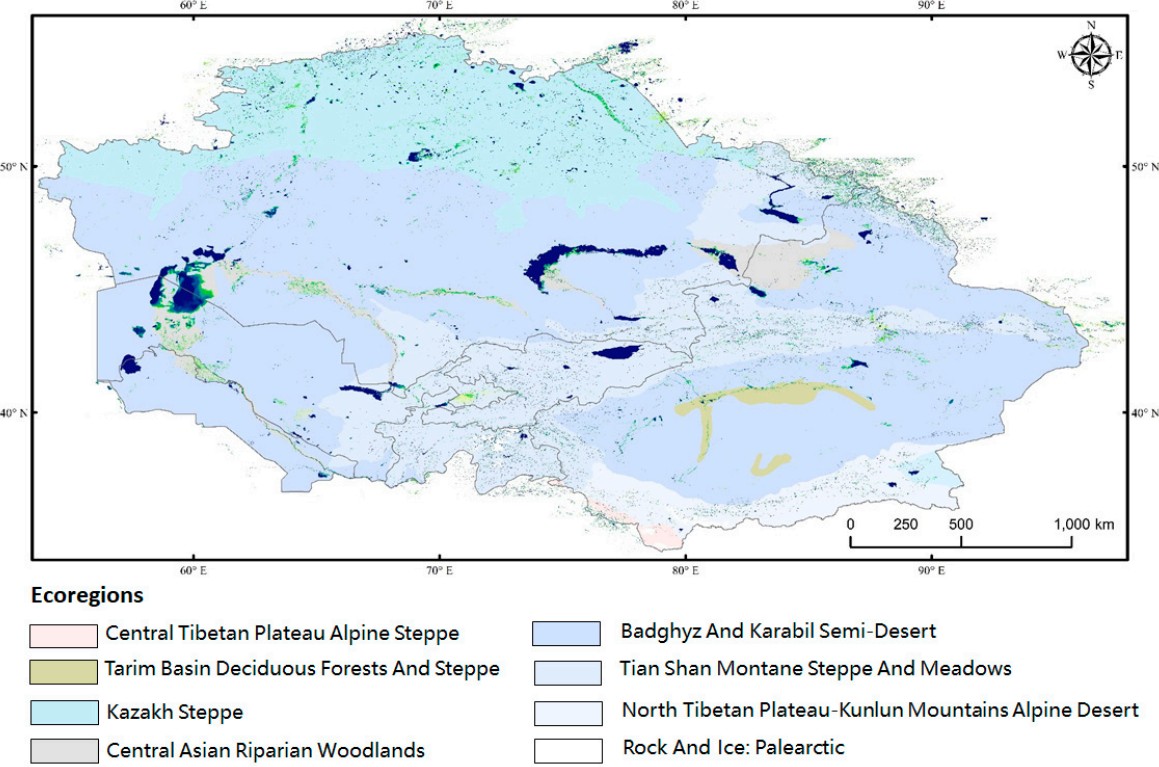

**Figure 7.** WOF distribution overlaid by eight terrestrial ecological zones in the arid and semi-arid regions of central Asia.

The maximum and minimum water bodies in the arid and semi-arid regions of central Asia are indicators of water variability, and were also qualitatively analyzed (Figure 8). The maximum extent of water in each year was mapped to represent the maximum water bodies, while the minimum water bodies (permanent water) were underwater throughout the year. The minimum water area accounted for 69.79% of the maximum water area on average. The area of non-seasonal, perennial water cover declined from 101,438.15 km$^2$ in 2000 to a minimum of 84,265.94 km$^2$ in 2014 and 85,702.31 km$^2$ in 2015. The maximum water area peaked at 155,408.62 km$^2$ in 2002, and dropped to the lowest value of 111,309.99 km$^2$ in 2014. There was a sharp drop in the maximum water area in 2011. There were linear decreasing trends for both the maximum and minimum bodies, but the decreasing rates (2825.7 km$^2$ per year) of maximum extent was almost double that of the minimum extent. However, the pattern of linear decrease for the minimum water bodies was more linearly significant when compared to that of the maximum bodies (R$^2$: 0.81 and 0.76).

The region showed strong seasonal variation of water cover (Figure 9). The water extent in the area reached its largest extent in April (119,663.47 km$^2$, SE = ±3192.18 km$^2$) and its smallest extent (105,574.20 km$^2$, SE = ±2015.45 km$^2$) in September. The water extents also showed a decreasing trend for each month over the 16 years, and the trend was closely linear (R$^2$ > 0.729). April showed the most significant decrease with nearly 2% shrinkage per year, while September showed the smallest reduction of 1.51% per year (Table 3).

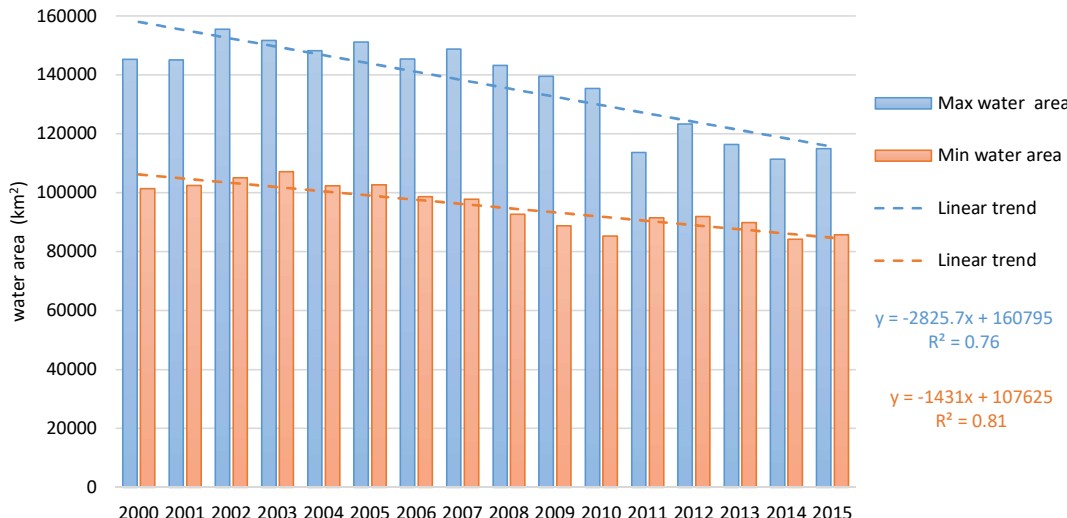

**Figure 8.** Changes to the maximum and minimum intra-annual water area from 2000 to 2015 in arid and semi-arid regions of central Asia.

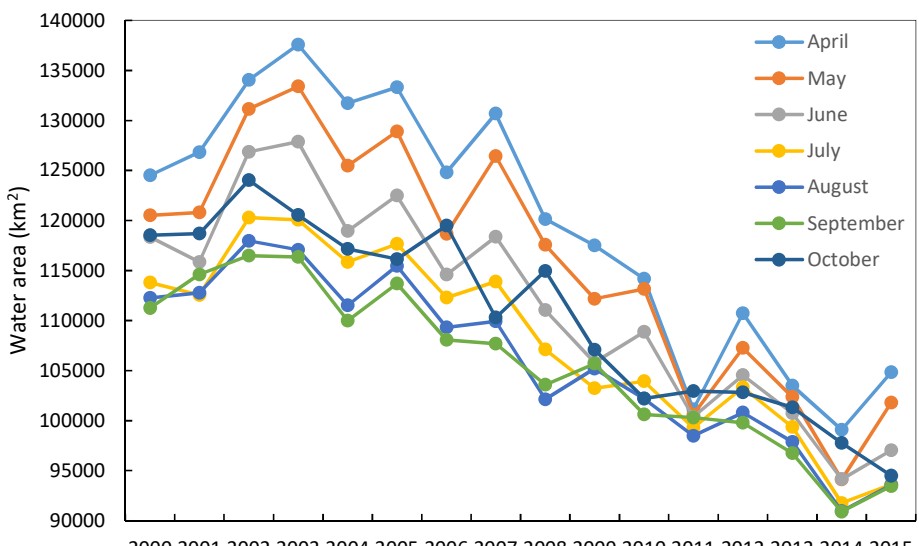

**Figure 9.** The changes in water area for each month across the year in the arid and semi-arid regions of central Asia.

**Table 3.** The statistics of the change pattern for Figure 9.

| Month | April | May | June | July | August | September | October |
|---|---|---|---|---|---|---|---|
| **AC/Slope** | −2317.39 | −2148.73 | −1957.20 | −1724.14 | −1631.15 | −1596.67 | −1828.94 |
| **RC** | −1.94 | −1.86 | −1.75 | −1.60 | −1.54 | −1.51 | −1.66 |
| **R2** | 0.747 | 0.729 | 0.794 | 0.814 | 0.856 | 0.889 | 0.880 |

AC/Slope: Absolute change (km$^2$/year); RC: Relative change (%/year).

## 4. Discussion

The water detection method has been assessed by comparing the derived global water map to the MODIS water dataset and national land cover datasets for the U.S. and Canada [1]. These static datasets are reliable references for assessing the accuracy of results for large regions or the globe, but are incapable of delineating the accuracy of the results for temporal surface water dynamics. Effort was made in this study to provide a more comprehensive assessment of the accuracy of water detection results for the arid and semi-arid regions of central Asia.

The overall accuracy of dynamic water detection (99.59%) in this study was comparable to that of Landsat water detection from Tulbure (99.9%), who employed a similar stratified random sampling strategy [16]. The results from Tulbure had a high omission error (12.8%) and low commission error (4.1%), however, there were larger commission errors (12.88%) than omission errors (1.76%) in this study. Although the time-series of the water records that were derived from the Landsat water identification method provided an accurate representation of the water distribution and dynamics in both the temporal and spatial dimension, the smaller omission errors typically occurred along rivers, small water bodies, and swamps where the presence of both water and vegetation within the pixel led to a failure to identify water, which can be further improved by applying sub-pixel and super-resolution mapping methods [18,53–55]. The larger commission errors were primarily located at the shaded areas in steep terrain, dense urban areas, or undetected cloud shadow, which can be seen in the northern part of Xinjiang (Altai Mountains). In addition, snow and ice are common sources of confusion due to their similar spectra, which means that the detected results from colder seasons are likely to overestimate the extent of water in locations covered by snow and ice. However, these known commission errors can be resolved with the lakes mask dataset if focusing on the analysis of lake dynamics [56].

The results also compared favorably to the global dataset developed by JRC [18]. In particular, the monthly dataset of this study had good spatio-temporal continuity due to temporal extension, which facilitates the time series analysis of water bodies. Figure 10 shows the comparison of the water distribution and dynamics between the interpolated results and JRC in 2015 for a region experiencing large changes. The first row in Figure 10 shows the water extent of the Tengiz-Korgalzhyn lake system in April 2015. The JRC monthly history occasionally showed some gaps and invalid observations, and was not able to capture the entire area of land during one month, thus reducing the accuracy of the monthly water detection. Using temporal interpolation, the synthesized monthly results were temporally comparable for every region, while the water detection for the JRC monthly history was possibly derived from two different dates over a month, especially obtained from the beginning and end of the month. In addition, the detection of some unobserved or contaminated pixels can be fixed for monthly interpolated results. The JRC Yearly Seasonality Classification collection contains a year-by-year classification of the seasonality of water based on the inundation values detected throughout the year. The third row in Figure 10 shows strong agreements have been found between the yearly results of the two datasets, especially in large and permanent water bodies, and the disagreements are mainly located in the small and seasonal water bodies, which indicates that the interpolated monthly water results are capable of depicting the dynamics throughout the year. Furthermore, the water extraction method has the ability to incorporate other sources of knowledge to further improve its accuracy. Although the reported accuracy was high in this study, other datasets (e.g., JRC dataset and regional water maps) can be used to further improve the accuracy in later studies.

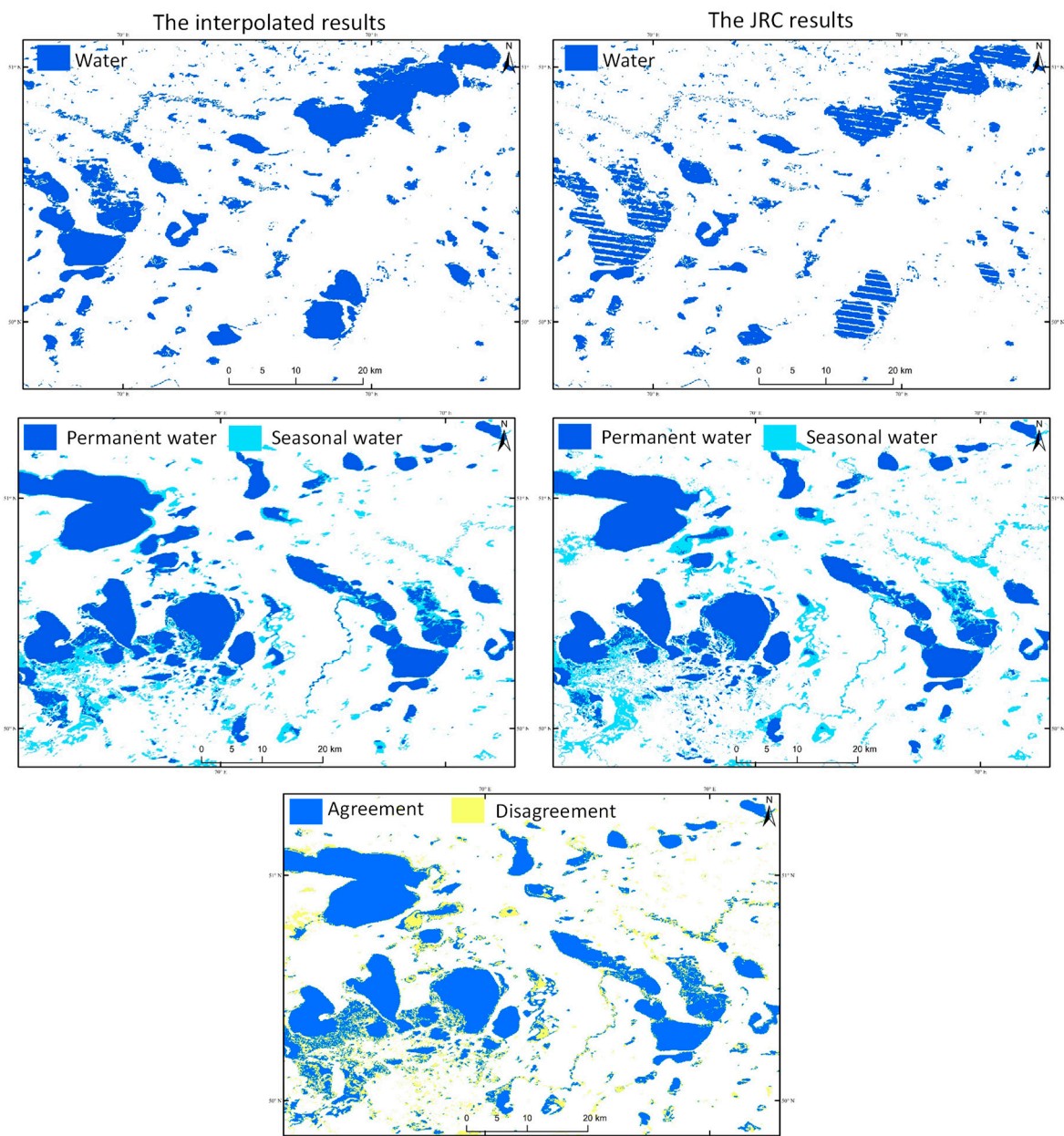

**Figure 10.** Comparison of the results (first column) to the JRC Yearly and Monthly Water Classification History (second column) as well as the agreement between their yearly datasets (third row) in 2015.

## 5. Conclusions

With water covers identified in each Landsat image acquired in the study area over the 16 years between 2000 and 2015, monthly water extents were derived by temporal interpolation over the derived multiple observations and inconsistent acquisition times of Landsat data availability. The analysis of surface water in the arid and semi-arid regions of central Asia showed a linear decreasing trend for the maximum and minimum water area, and the water area was linearly reduced each month with strong seasonality. A statistically rigorous accuracy assessment was performed to validate the surface water based on a stratified sampling design to provide unbiased accuracy estimates. The accuracy of the water results was high and acceptable over time, especially for warmer seasons. It is an imperative step to quantify the uncertainty of the accuracy assessment and provide a guide to the use of the method for the trend of surface-water dynamics over central Asia or other regions and globally, and its response to environment change needs to be further examined.

**Author Contributions:** Conceptualization, M.F. and J.S.; Methodology, X.C.; Software, S.C.; Validation, X.C. and Q.Y.; Formal analysis, M.F.; Investigation, Q.S.; Resources, J.S.; Data curation, X.C.; Writing—original draft preparation, X.C.; Writing—review and editing, M.F. and S.C.; Visualization, Q.S.; Supervision, Y.W. and J.L.; Project administration, J.S.; Funding acquisition, S.C.

**Funding:** This research was funded by the Strategic Priority Research Program of the Chinese Academy of Sciences (Grant No. XDA20100104) and the Basic Scientific Research Operating Expenses of the Chinese Academy of Surveying and Mapping (No. 20603020004096004 AR1904).

**Acknowledgments:** The authors are indebted to the National Aeronautics and Space Administration for providing the MODIS data (https://modis.gsfc.nasa.gov/), U.S. Geological Survey server (http://glovis.usgs.gov), and Global Land Cover Facility server (http://www.landcover.org) used in this study. In addition, we would like to thank the anonymous reviewers for their helpful comments and suggestions in enhancing this manuscript.

**Conflicts of Interest:** The authors declare no conflict of interest.

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
