# Peer review of "Landsat-Based Estimation of Seasonal Water Cover and Change in Arid and Semi-Arid Central Asia (2000–2015)"

_remotesensing, doi:10.3390/rs11111323_

Round 1

Reviewer 1 Report

The paper exercises the extraction of water bodies from temporal imagery. However, there is no novelty from either technical or application point of view. Moreover, the validation approach is not scientifically sound. Because the reference data is a global product which is not suitable for assessing the authors’ objective that they wanted to produce a more reliable map at the regional scale. I am not recommending the publication of this work. Here are my comments:

Lines 68-69: is this really true that there is no temporal study in central Asia about the surface water extent? What about other regions? You should provide a novel theory/application and repeating the previous studies make no sense.

Line 110-111: how do you use a low resolution product (250 m) to validate a higher resolution map (30 m)? Usually the opposite can be done.

Again, if you are taking JCR data set as reference, why you just don’t use them? You claimed before that your method will be region-specific without the typical uncertainties of global products.

Lines 145-150: it is unclear how this step is working and what the purpose is.

State of the art methods for extraction of water bodies are not addressed in this study. For example sub-pixel mapping methods:

Super resolution mapping of the waterline from remotely sensed data

Reconstruction of River Boundaries at Sub-Pixel Resolution: Estimation and Spatial Allocation of Water Fractions

Super-Resolution Mapping o fWetland Inundation from Remote Sensing Imagery Based on Integration of Back-Propagation Neural Network and Genetic Algorithm

Author Response

Dear reviewer,

Thank you for reviewing our manuscript “Landsat-based estimation of seasonal water cover and change in arid and semi-arid central Asia (2000 – 2015)” (ID: remotesensing-475003) and your comments. We have made the following modifications, which we hope to meet with approval.

Yours sincerely,

Xianghong Che

Reviewer 1

Comment 1: Lines 68-69: is this really true that there is no temporal study in central Asia about the surface water extent? What about other regions? You should provide a novel theory/application and repeating the previous studies make no sense.

Response: The sentence at Line 68-69: “there are no regular time-series and medium/high geospatial datasets to study water surface changes in the region” has been modified to “Satellite-based studies of lake dynamics in central Asia have been limited to specific lakes and/or irregular time periods [16, 22], with little knowledge of the accuracy.” at Line 84 – 85, and then the novel application has been followed at Line 85 – 88.

Comment 2: Line 110-111: how do you use a low resolution product (250 m) to validate a higher resolution map (30 m)? Usually the opposite can be done. Again, if you are taking JRC data set as reference, why you just don’t use them? You claimed before that your method will be region-specific without the typical uncertainties of global products.

Response: In this study, the 250-m resolution MODIS water dataset (MOD44W) was not used for validation, instead it was only used for creating the sampling strata for building the water classification model (the detail is discussed in Feng et al. section 2.2.2). The water extracting model for each Landsat image is built mainly by taking the information from the Landsat image itself (e.g., the water sensitive bands, and water indices), and it also incorporates other sources of information (e.g., the MOD44W dataset) to reduce the uncertainty in its predictions. The JRC dataset could be incorporated to further improve the accuracy, as well as other regional or globally available datasets. We have added inserted a sentence at Line 313-316 in the discussion to address the point.

Comment 3: Lines 145-150: it is unclear how this step is working and what the purpose is.

Response: The step is to create temporal consistent monthly water frequency record for each location. As presented in previous subsections, water is identified in each Landsat image, and the time-series of Landsat-derived records can be extracted from the stack of Landsat water datasets. However, the density of the valid Landsat observations for each location varies over time due to many factors, including the revisiting dates of Landsat satellites, atmospheric condition, etc. The step calculates water frequency for each month using all the Landsat-derived records in the month, and the months with no valid observations are estimated using interpolation to produce a continuous time-series water record.

Comment 4: State of the art methods for extraction of water bodies are not addressed in this study. For example sub-pixel mapping methods: Super resolution mapping of the waterline from remotely sensed data; Reconstruction of River Boundaries at Sub-Pixel Resolution: Estimation and Spatial Allocation of Water Fractions; Super-Resolution Mapping of Wetland Inundation from Remote Sensing Imagery Based on Integration of Back-Propagation Neural Network and Genetic Algorithm.

Response: This study focuses on deriving time-series water records from Landsat images. We believe that sub-pixel mapping and super-resolution mapping of water can be applied to further refine the results from this study, especially for the small water bodies. A sentence has been inserted to the discussion to address the point at Line 286 - 291.

Reviewer 2 Report

Dear Authors,

The submitted work is well designed and implemented but i cannot see something new - what is the difference of it from this one which offer global cover from 1984 onward? > https://global-surface-water.appspot.com/.

Thus, i cannot accept it for publication at the Remote Sensing journal.

Author Response

Dear reviewer,

Thank you for reviewing our manuscript “Landsat-based estimation of seasonal water cover and change in arid and semi-arid central Asia (2000 – 2015)” (ID: remotesensing-475003) and your comments. We have made the following modifications, which we hope to meet with approval.

Yours sincerely,

Xianghong Che

Reviewer 2

Comment 1: The submitted work is well designed and implemented but i cannot see something new - what is the difference of it from this one which offer global cover from 1984 onward? > https://global-surface-water.appspot.com/.

Response: We appreciate the comments! As the reviewer pointed out, both the Joint Research Centre (JRC) water datasets and the datasets presented in this paper were derived from Landsat images to provide water distribution over space and temporal dimensions. However, in addition to its global calibration, the JRC dataset has been found to show quality issues in the region and temporal discontinuity issues (as the examples below). The paper is to derive a completed time-series of water cover records from Landsat images in the region, conduct a validation of the dataset’s accuracy, and report the water distribution and dynamics in the region.

Figure 10. Comparisons of results (first column) to JRC Yearly and Monthly Water Classification History (second column) in 2015

Reviewer 3 Report

2019.  

This is a very competent manuscript. It is well written and has a logical organization. The tables and figures are professional and informative. There is an excellent set of appropriate references. 

The authors are to be congratulated on the very large amount of Landsat and other imagery acquired and processed.  It is an excellent example of the amount of increased science that can be accomplished with the availability of free imagery.  The topic will be of limited interest to the remote sensing community other than the large volume of data.  It will be of greater interest to applied scientists with an interest in central Asia and changing water resources.

As in almost any manuscript, there are editorial suggestions for consideration by the authors, several of which follow:

1.      It varies by scientist and journal but personal pronouns (we, our) are not typical in technical text.

2.      There is inconsistent use of serial commas.

3.      Line 56, perhaps their for its?

4.      Line 58, to map or for mapping?

5.      Line 71, why first observations?  Will there be others?

6.      Most figure captions do not end with a period, but some do.

7.      Figure 6 subscenes might have a scale bar.  Also Figure 10.

8.      Line 228, trends?

9.      Line 230, delete than.

10.  Line 271, of for on?

11.  Line 275, dates?

12.  There is inconsistency in the format of references.  Some article titles are in upper case, some journal titles in lower case, and one journal name is shortened.

As stated, this is a very good manuscript and certainly appropriate for Remote Sensing.

Author Response

Dear reviewer,

Thank you for reviewing our manuscript “Landsat-based estimation of seasonal water cover and change in arid and semi-arid central Asia (2000 – 2015)” (ID: remotesensing-475003) and your comments. We have made the following modifications, which we hope to meet with approval.

Yours sincerely,

Xianghong Che

Reviewer 3

Comment 1: It varies by scientist and journal but personal pronouns (we, our) are not typical in technical text.

Response: The terms “we” or “our” has been removed through the paper as the reviewer suggests.

Comment 2: There is inconsistent use of serial commas.

Response: Inconsistent use of serial commas has been removed though the paper.

Comment 3: Line 56, perhaps their for its?

Response: The term “its” has been modified to “their” at Line 57 as the reviewer suggests.

Comment 4: Line 58, to map or for mapping?

Response: The term “to mapping” has been modified to “to map” at Line 59 as the reviewer suggests.

Comment 5: Line 71, why first observations? Will there be others?

Response: The sentence “Here we describe temporal extension of the algorithm and first observations from its application in central Asia.” has been modified to “Here the study describes temporal extension of the algorithm and its first application in central Asia” at Line 88. The method could be further developed for regional and global applications.

Comment 6: Most figure captions do not end with a period, but some do.

Response: All of figure captions through the paper has been modified to end with a period.

Comment 7: Figure 6 sub-scenes might have a scale bar. Also Figure 10.

Response: The scale bars have been added for each sub-scene of Figure 6. The scale bars of Figure 10 lied out on the bottom of each sub-scene.

Comment 8: Line 228, trends?

Response: The term “trend” at Line 228 has been modified to “trends” at Line 256 as the reviewer suggests.

Comment 9: Line 230, delete than.

Response: The term “than” at Line 258 has been removed.

Comment 10: Line 271, of for on?

Response: The term “in April on 2015” has been modified to “in April of 2015” at Line 302.

Comment 11: Line 275, dates?

Response: The term “date” has been modified to be plural format “dates” at Line 306.

Comment 12: There is inconsistency in the format of references. Some article titles are in upper case, some journal titles in lower case, and one journal name is shortened.

Response: As the review suggests, the first letters of all article titles and each word for journal titles have been capitalized to ensure consistent format in the reference. The shortened journal name in the reference [40] has been modified at Line 429.

Reviewer 4 Report

It is a very interesting study dealing with the use of a big series of Landsat data to estimate the seasonal water cover and change in Central Asia.

The processing and classification demonstrate and reveal a good methodology and a very powerful system of servers to manipulate the enormous size of the input data.

Nevertheless, the Introduction is very weak and needs a more detailed analysis as far as it concerns the existence of other studies using Landsat data in such cases and in the demonstration of other kind of methodologies (using of course the appropriate references).

Moreover, a more thorough analysis of the Methodology is necessary to be made. Lots of points are just referenced without any further analysis or clarification.

Finally, the quality improvement of some figures is needed

(see the attached comment file).

Author Response

Dear reviewer,

Thank you for reviewing our manuscript “Landsat-based estimation of seasonal water cover and change in arid and semi-arid central Asia (2000 – 2015)” (ID: remotesensing-475003) and your comments. We have made the following modifications, which we hope to meet with approval.

Yours sincerely,

Xianghong Che

Reviewer 4

Comment 1: the Introduction is very weak and needs a more detailed analysis as far as it concerns the existence of other studies using Landsat data in such cases and in the demonstration of other kind of methodologies (using of course the appropriate references).

Response: We has updated the Introduction section to expand the summary of existing studies at Line 59-83, and included relevant references:

[15]  Tulbure M.G., Broich M., Stehman S.V., et al. Surface water extent dynamics from three decades of seasonally continuous Landsat time series at subcontinental scale in a semi-arid region [J]. Remote Sensing of Environment, 2016, 178: 142-157.

[16]  Mueller N., Lewis A., Roberts D., et al. Water observations from space: Mapping surface water from 25 years of Landsat imagery across Australia [J]. Remote Sensing of Environment, 2016, 174: 341-352.

[17]  Halabisky M., Moskal L.M., Gillespie A., et al. Reconstructing semi-arid wetland surface water dynamics through spectral mixture analysis of a time series of Landsat satellite images (1984–2011) [J]. Remote Sensing of Environment, 2016, 177: 171-183.

Comment 2: A more thorough analysis of the Methodology is necessary to be made. Lots of points are just referenced without any further analysis or clarification. Page 5 – Line 132: I think you should give a short analysis of the algorithm instead of just give a reference.

Response: This paper focuses on producing time-series water records from Landsat images, reporting its accuracy, and analyzing the spatio-temporal dynamic in the region. Because the water extraction method has been published already [1], this paper only provides a short description of it at line 147-157.

[1] Feng M., Sexton J.O., Channan S., et al. A global, high-resolution (30-m) inland water body dataset for 2000: first results of a topographical-spectral classification algorithm [J]. International Journal of Digital Earth, 2016, 9(2): 113-133.

Comment 3: Page 2 - Line 50: Delete Moderate Resolution

Response: The term “Moderate Resolution” was repeated and has been removed at Line 52.

Comment 4: Page 2 - Line 52: You probably mean small changes

Response: The term “large changes” has been modified to “small lakes” at Line 57.

Comment 5: Page 2 - Line 64: From 1984 to 2015

Response: This sentence has been modified to “Pekel et al. [18] retrieved a high-resolution mapping dataset of global surface water quantifying changes in global surface water over the past 32 years from 1984 to 2015 at 30-m resolution using the entire archive of the Landsat 5 TM, the Landsat 7 ETM+ and the Landsat 8 OLI images.” at Line 77 -80.

Comment 6: Page 2 - Line 77: study area or area

Response: The term “in the study” has been modified to “in the study area” at Line 94.

Comment 7: Page 3 – Line 86: Figure 1. You should use a better-quality background image

Response: Figure 1 has been modified to present a higher quality image.

Comment 8: Page 3 – line 90: You should rephrase this sentence (to be clearer)

Response: This sentence at Line 90 has been modified to “The Level-1 Terrain Corrected (L1T) Landsat images between 2000 and 2015 with the cloud cover threshold of < 80% were selected. The study area is covered by 315 Landsat World Reference System (WRS)-2 scenes, and 70,186 Landsat images were downloaded from USGS/EROS (http://landsat.usgs.gov/) (Figure 2) at Line 114 – 116.

Comment 9: Page 4 – Line 101: Figure 2. You should increase the quality of this figure's legend

Response: The quality of Figure 2's legend has been improved as the reviewer requests.

Comment 10: Page 4 – Line 103: You should probably use more references (eg. https://pubs.er.usgs.gov/publication/ofr20131057)

Response: Another relevant reference has been added for LEDAPS algorithm at Line 128 as the reviewer requests.

[30] Schmidt G., Jenkerson C.B., Masek J., et al., Landsat ecosystem disturbance adaptive processing system (LEDAPS) algorithm description [R]: US Geological Survey, 2013

Comment 11: Page 4 – Line 105: You should probably use more references (eg. https://pubs.er.usgs.gov/publication/ofr20131057)

Response: Two relevant references has been added for LaSRC algorithm as the reviewer requests at Line 129.

[30] Schmidt G., Jenkerson C.B., Masek J., et al., Landsat ecosystem disturbance adaptive processing system (LEDAPS) algorithm description [R]: US Geological Survey, 2013.

[31]  Claverie M., Ju J., Masek J.G., et al. The harmonized Landsat and Sentinel-2 surface reflectance data set [J]. Remote Sensing of Environment, 2018, 219: 145-161.

Comment 12: Page 4 – Line 126: You should explain which indices you used and their equations

(adding of course and the relevant references)

Response: The indices at Line 150 were used as inputs for building the water classification model, and the detail of indices and their equations are discussed in Feng et al. section 2.2.2. This comment has been addressed at the response to Comment 2.

Comment 13: Page 6 – Line 172: Visual examining...Why didn't you utilize a geospatial approach?

Response: The work is to collect high-accuracy dataset for validating the spatio-temporal water dataset derived from Landsat. The visual interpretations were conducted by remote sensing experts to provide an independent and relatively higher quality reference dataset for the validation.

Comment 14: Page 6 – Line 175: The methodology used for the classification of water needs further

Analysis

Response: The monthly water records at Line 168 were produced from temporal aggregating water layers identified in each Landsat image. The aggregation method was described in section 2.3.2, and the Landsat water identification method was described in 2.3.1. No modification was made to address this comment.

Comment 15: Page 6 – Line 180: Figure 5. It would be better if you choose the blue color for the

Permanent water, a cyan color for seasonal water and brown color for permanent land

Response: Permanent water, seasonal water and permanent land has been colored with blue, cyan and brown, respectively, as the reviewer requests for Figure 5 at Line 201.

Comment 16: Page 9 – Line 218: Figure 7. Provide a Figure of better quality and analysis.

Response: The quality of Figure 7 has been improved as the reviewer requests. The analysis of Figure 7 has been added at Line 237– 240.

Comment 17: Page 10 – Line 243: Figure 9. It would be interesting if you could clarify the reason of this Water Area shrink during the last 15 years.

Response: We agree with this. It will require further investigated in the future to identify the driving forces of the area decreasing during the last 15 years.

Comment 18: Page 11 – Line 268: You mean by visual interpretation? You should better use a geoprocessing tool to make the comparison

Response: The comparison was to reveal the quality issues in the JRC and our datasets, and we believe that visual comparison can provide intuitive results for demonstrating the quality issues.

Reviewer 5 Report

The manuscript by Che et al focuses on mapping surface water over central Asia over a fifteen year period of 2000-2015 using Big Data technologies and open source Landsat data archive. The study uses Machine Learning algorithm to achieve this with excellent accuracy. As surface water is an important aspect of ecosystems and economics, being able to develop accurate remote sensing techniques is of crucial importance. Global datasets created recently do not have regional focus and hence studies of this nature could help improve our understanding of surface water dynamics at regional scale.

The manuscript presents the study clearly with detailed methodology etc. However, why this study chose the particular study location was not stated and hence needs to be added.

The study is a contribution to the surface water mapping area and should be considered for publication in the present form.

Author Response

Dear reviewer,

Thank you for reviewing our manuscript “Landsat-based estimation of seasonal water cover and change in arid and semi-arid central Asia (2000 – 2015)” (ID: remotesensing-475003) and your comments. We have made the following modifications, which we hope to meet with approval.

Yours sincerely,

Xianghong Che

Reviewer 5

Comment 1: The manuscript presents the study clearly with detailed methodology etc. However, why this study chose the particular study location was not stated and hence needs to be added.

Response: We appreciate the comment! The arid and semi-arid area is sensitive to water availability and is also known for its dramatic changes of water in the past decades. We have modified the manuscript to address the comment in section 2.1 at Line 102 – 107:

Although water bodies in this region are of high ecological and economic importance, significant changes had occurred in the last decades, thus leading to land desertification, salinization, degradation of vegetation, and biodiversity loss [3, 12]. Accurate datasets and time-series records of water bodies are helpful information for not only expanding our understanding of natural variability and human interaction, but also addressing these ecosystem and environmental issues.

[3] Bai J., Chen X., Li J., et al. Changes in the area of inland lakes in arid regions of central Asia during the past 30 years [J]. Environmental Monitoring and Assessment, 2011, 178(1-4): 247-256.

[12] Klein I., Dietz A.J., Gessner U., et al. Evaluation of seasonal water body extents in Central Asia over the past 27 years derived from medium-resolution remote sensing data [J]. International Journal of Applied Earth Observation and Geoinformation, 2014, 26: 335-349.

Round 2

Reviewer 1 Report

The revised manuscript does not show any significant improvement with respect to the previous submission. The authors didn't reply that what is the novelty of this work. The work is behind the state of the art and does not provide any technical or application novelty. Moreover, the design of experiments and validation is still vague. I am not recommending the publication of this work as it is critical to keep high the level of papers in Remote Sensing.

Author Response

Dear reviewer,

Thank you for reviewing our manuscript “Landsat-based estimation of seasonal water cover and change in arid and semi-arid central Asia (2000 – 2015)” (ID: remotesensing-475003) and your comments. We have made the following modifications, which we hope to meet with approval.

Yours sincerely,

Xianghong Che

Reviewer 2 Report

Dear authors,

The revised version along with the provided responses clarify more the situation.

In figure 10, second row, can you make a difference map (left minus right) to see the differences? I don't ask for the above row as the striped LT7 data will show differences.

And the dif map to be added below the second row.

Looking forward

Author Response

Dear reviewer,

Thank you for reviewing our manuscript “Lanndsat-based estimation of seasonal water cover and change in arid and semi-arid central Asia (2000 – 2015)” (ID: remotesensing-475003) and your comments. We have made the following modifications, which we hope to meet with approval.

Yours sincerely,

Xianghong Che

Reviewer 2

Comment 1: The revised version along with the provided responses clarify more the situation.

In figure 10, second row, can you make a difference map (left minus right) to see the differences? I don't ask for the above row as the striped LT7 data will show differences. And the dif map to be added below the second row.

Response: We appreciate the comment! The agreement and disagreement map between interpolated and JRC yearly datasets have been added as in Figure 10 as the review suggested. The analysis of the difference have been inserted at the Line 310-312.
